# Genetic diversity of the two-spotted stink bug *Bathycoelia distincta* (Pentatomidae) associated with macadamia orchards in South Africa

**Elisa Pal**[1], **Jeremy D. Allison**[1,2], **Brett P. Hurley**[1], **Bernard Slippers**[3], **Gerda Fourie**[3]*

**1** Department of Zoology and Entomology, Forestry and Agricultural Biotechnology Institute (FABI), University of Pretoria, Pretoria, South Africa, **2** Natural Resources Canada-Canadian Forest Service, Great Lakes Forestry Centre, Sault Ste. Marie, Canada, **3** Department of Biochemistry, Genetics and Microbiology, Forestry and Agricultural Biotechnology Institute (FABI), University of Pretoria, Pretoria, South Africa

* gerda1.fourie@fabi.up.ac.za

**Data Availability Statement:** Sequences of COI (accession numbers: OM263477-OM263633), and Cytb (accession numbers: OM219650-OM219806)

## Abstract

The South African macadamia industry is severely affected by a complex of stink bugs, dominated by the two-spotted stink bug, *Bathycoelia distincta* Distant (Pentatomidae). This species was first discovered during the spring of 1984 in the Limpopo province. Although considerable effort has been spent trying to manage this pest, it continues to be a pest of concern for the macadamia industry. Information on the genetic diversity of this species is lacking, despite the potential relevance of such information for management strategies. The present study aimed to characterise the genetic diversity of *B. distincta* populations in South Africa. The Cytochrome c Oxidase Subunit 1 (COI) and cytochrome b (Cytb) gene regions were sequenced from individuals collected from the three main regions of macadamia production over three different seasons (2018–2020). An overall high haplotype diversity (COI = 0.744, Cytb = 0.549 and COI+Cytb = 0.875) was observed. Pairwise mean genetic distance between populations from each region varied from 0.2–0.4% in both datasets, which suggests the absence of cryptic species. The median joining network for both datasets consisted of one or two central haplotypes shared between the regions in addition to unique haplotypes observed in each region. Finally, low genetic differentiation ($F_{ST}$ < 0.1), high gene flow (Nm > 1) and the absence of a correlation between genetic and geographic distance were estimated among populations. Overall, these results suggest that the *B. distincta* populations are not structured among the areas of macadamia production in South Africa. This might be due to its ability to feed and reproduce on various plants and its high dispersal (airborne) between the different growing regions of the country along with the rapid expansion of macadamia plantations in South Africa.

of B. distincta were deposited in NCBI GenBank (www.ncbi.nlm.nih.gov). Sequences of COI were also deposited to BOLD (http://boldsystems.org: PMSL01-66 sequences from Limpopo; PMSM01-45 sequences from Mpumalanga; PMSK01-46 sequences from Kwazulu-Natal.

**Funding:** We would like to acknowledge the University of Pretoria, the Forestry and Agricultural Biotechnology Institute (FABI), the Centre for Excellence in Plant Biotechnology (CPHB), Macadamia South Africa NPC (SAMAC) and NRF Thuthuka for financial support. There was no additional external funding received for this study.

**Competing interests:** The authors have declared that no competing interests exist.

# Introduction

The Pentatomidae is one of the largest families within the Heteroptera with more than 4,700 species distributed worldwide [1]. Phytophagous and polyphagous, Pentatomidae are a major concern to agricultural production around the world, including nut crops [2, 3]. In nut crops, stink bugs can cause direct feeding damage (e.g., nut abortion, discolouration, fruit drop) by insertion of their mouthparts into developing nuts inducing losses in yield and kernel nut quality [4–6]. Stink bugs can also cause indirect feeding damage by transmission of pathogens. This has been demonstrated especially in *Nezara viridula* (L.) with the transmission of *Pantoea agglomerans* into cotton bolls [7, 8].

In South Africa, several nuts are produced and exported worldwide, including macadamia nuts. Considered as the current world's largest producer of macadamia nuts, the industry is continuously growing with about 5000 ha of trees planted annually [9]. However, this production is severely affected by various Heteroptera from the families Coreidae and Pentatomidae [10–12], causing approximately 15.23 million USD in losses due to nut damage annually [12, 13]. Among the different stink bug species occurring in macadamia orchards, *Bathycoelia distincta* Distant (Heteroptera: Pentatomidae) is the most abundant [10, 14, 15] and one of the most damaging pests for the industry due to its long proboscis which can cause kernel damage throughout the entire season [11, 16]. *Bathycoelia distincta* was discovered during the spring of 1984 in the Limpopo province and is now present in plantations in all macadamia production regions. Nymphs and adults have been recovered in the orchards even when no or few nuts are available on trees [10, 14], suggesting that this pest is breeding in the orchards and well established in all production areas in the country.

Control of *B. distincta* and other stink bug species has relied mainly on chemical insecticides. However, the prevalence of insecticide resistance and outbreaks of secondary pests coupled with increased environmental concerns have led to more integrated pest control approaches [17–19]. Some studies have been conducted to understand the seasonal occurrence [14] and the distribution patterns of *B. distincta* [20] to help with the implementation of IPM methods in South Africa [21]. Nevertheless, a comprehensive understanding of *B. distincta* population dynamics is lacking and required for the design of effective management strategies.

Population genetic studies can provide knowledge about the origin, migration patterns, genetic structure, and dynamics of pest populations. Exploring the genetic variability of a species among regional populations is also crucial to detect the existence of cryptic species complexes that could directly affect the efficiency of pest control efforts [22, 23]. Indeed, the utilisation of pheromones is often species-specific and may not work if subpopulations or cryptic species occur in the same geographical area, as has been demonstrated in various lepidopteran species [24–26]. Similarly, successful biological control depends on the use of species-specific parasitoid wasps. For example, a recent study discovered that cryptic species of the parasitoid *Ganaspis brasiliensis* have different affinities towards their hosts regardless of their food source, and as such impact the biological control of *Drosophila suzukii* (Matsumura) [27]. Another study on generalist parasitoids from the subfamily Aphidiinae revealed the presence of multiple cryptic species which are each in fact associated with a different host species [28]. Host preferences also exist in parasitoids of the Pentatomidae [29]. For example, *Trissolocus utahensis* (Ashmead) parasitizes more eggs of *Chlorochroa uhleri* (Stål) than *Chlorochroa sayi* (Stål) [30].

Population genetic studies have also highlighted genetic differentiation which may occur between insecticide resistant and susceptible populations. This may be a result of insecticide selection [31–33], management practices [34, 35], or other factors such as species isolation [36, 37]. Insecticide resistance has been reported for several Pentatomidae species [38]. For

example, *Euschistus heros* (Fabricius) populations in Brazil have shown reduced susceptibility to various insecticides such as organophosphates and endosulfan [38, 39]. Concern regarding the reduced susceptibility of *E. heros* led to a large geographical survey and geostatistical analysis of bioassays with various insecticide formulations in order to map and identify areas with high risk of insecticide control failure [40].

The utilisation of mitochondrial DNA (mtDNA) has become an effective method to detect genetic variation among regional populations [41–44] due to its central role in metabolism and its high conservation rate across species in some sites, but high mutation rate in other sites to allow for species delimitation [41–43]. Although this method has been criticized [45–48], population genetic studies have been widely used among insect taxa in a variety of disciplines, including general ecology and evolution [49], to inform and improve management strategies. In addition, the mitochondrial Cytochrome c Oxidase Subunit 1 (CO1) gene region has been used to determine intraspecific divergence rates. To this end intraspecific divergence of 3 to 5% in Heteroptera has been found to delineate cryptic species within populations [50–53] in comparison to 2% intraspecific divergence rates for some other insect groups [42].

Genetic diversity of Pentatomidae species has mostly been studied using the COI gene region [50–52, 54, 55], although additional genes encoded by the mitochondria such as the cytochrome b (Cytb) are also commonly used [49, 56]. The occurrence and genetic diversity of *Halyomorpha halys* (Stål) has been widely studied in several countries [57–62] and such information is also available for *N. viridula* [63–65]. For example, in Brazil, various studies examined population structure and variation among populations of *N. viridula* [66, 67], *E. heros* [68, 69] and *Loxa spp.* [70] from different geographic regions. The genetic variability of *Chinavia hilaris* (Say), *C. uhleri*, *C. sayi*, and *Thyanta pallidovirens* (Stål) has also been examined to determine the presence of cryptic species in order to assist with pest management in pistachio orchards [53].

The present study aimed to investigate the genetic diversity of *Bathycoelia distincta* in South Africa. We examined the COI and Cytb gene regions from individuals collected from different regions where macadamia is planted commercially. We determined the genetic variability of *B. distincta* populations within and between the three main macadamia production areas of South Africa. We also consider whether there is any evidence of cryptic species.

## Materials and methods

### Ethics statement

No endangered or protected species were involved in this study. No national permissions were required for this study. All work on this project was conducted with permission from landowners.

### Sample collection

Stink bugs were collected from different macadamia farms in Limpopo, Mpumalanga, and Kwazulu-Natal from October 2017 to March 2020 (Tables 1 and 2). Insects were collected as adults, from macadamia trees after insecticide application using a beating cloth under the trees. Samples were preserved in ethanol (> 95%) at -20°C until molecular analysis.

The two-spotted stink bug specimens were identified based on external morphological features described in literature [71] and also confirmed morphologically by an entomologist at the Agricultural Research Council—Plant Health and Protection (Pretoria, South Africa). Insects used for this study are conserved in our collection located at University of Pretoria. Vouchered specimens were pinned, accession number assigned (PENT00026-PENT00030)

**Table 1. Sampling locations where *B. distincta* was collected between October 2017 to March 2020 in South Africa provinces.**

| Location | n[1] | Latitude (S) | Longitude (E) |
|---|---|---|---|
| Farm L1 | 38 | 23˚04'41.1" | 30˚08'31.1" |
| Farm L2 | 28 | 23˚05'34.6" | 30˚14'23.3" |
| Farm M1 | 45 | 25˚04'50.4" | 31˚01'08.4" |
| Farm K1 | 46 | 31˚01'41.2" | 30˚13'13.1" |

[1] Number of specimens collected and used in this study to determine genetic diversity.

**Table 2. Distance in straight line (Km) among the different sampling sites of *Bathycoelia distincta* population.**

| Distance (Km) in straight line | Farm L1 | Farm L2 | Farm M1 | Farm K1 |
|---|---|---|---|---|
| Farm L1 | - | - | - | - |
| Farm L2 | 10 | - | - | - |
| Farm M1 | 242 | 234 | - | - |
| Farm K1 | 885 | 881 | 667 | - |

and deposited in the National Collection of Insects located at the Agricultural Research Council—Plant Health and Protection (Pretoria, South Africa).

## DNA extraction, PCR amplification and sequencing

The genetic diversity of *B. distincta* was determined by analysing sequence data from sections of the mitochondrial COI and Cytb genes. DNA was extracted from leg tissue of 157 adult specimens (Table 1) using the NucleoSpin® DNA insect (Macherey-Nagel GmbH & Co. KG, Düren, Germany) kit, following the manufacturer's protocol for tissue extraction. A negative control was also carried out with all the kit solutions but without insect tissue to check for contamination. The DNA quantity was measured using the Thermo Scientific NanoDrop® ND-1000 spectrophotometer (Wilmington, DE, USA). The quantity of DNA in samples ranged from 20–40 ng/µl.

The COI gene region was amplified using the universal forward LCO1490 (5′–GGTCAA-CAAA TCATAAAGATATTGG-3′) and reverse HCO2198 (5′–TAAACTTCAGGGTGACCA AAAAATCA-3′) primer [72], and the Cytb gene region was amplified using the forward (5′–GGATATGTTTTACCTTGAGGACA-3′) and the reverse (5′–GGAATTGATCGTAA-GATTGCGTA-3′) primer [66, 73]. Polymerase Chain Reactions (PCR) were performed in a total volume of 25 µL. For amplification of the COI gene, each PCR contained 5 µL of 5X PCR Buffer (dNTPs and MgCl2 included), 0.5 µL Taq DNA Polymerase (Bioline, South Africa), 0.5 µL of each primer (10 mM), 16.5 µL of distilled water and 2 µL of 50–100 ng DNA. PCR cycling conditions consisted of denaturation at 94˚C for 1 min, followed by five cycles with denaturation at 94˚C for 1 min, annealing at 45˚C for 90 sec, and extension at 72˚C for 90 sec, followed by another 30 cycles with denaturation at 94˚C for 1 min, annealing at 50˚C for 90 sec, extension at 72˚C for 1.30 min, and final extension at 72˚C for 5 min. The amplification of the Cytb gene was performed with 50–100 ng of DNA, 1.5 µL of MgCl2, 1 µL of dNTPs (10 µM), 1 µL of each primer (10 mM), 2.5 µL of 10X PCR buffer, and 0.5 µL of FastStart Taq DNA Polymerase (Roche, Molecular Biochemicals, Manheim, Germany). The PCR cycles consisted of initial denaturation at 94˚C for 5 min, followed by 35 cycles of denaturation at 94˚C for 45 sec, annealing at 50˚C for 30 sec, extension at 72˚C for 2 min and final extension at

72˚C for 10 min. With each run, negative controls without DNA in the PCR reaction were performed for PCR validation.

The PCR products were verified on agarose gel (1.5% w/v) with BioRad Gel Doc™ Ez Imager and then purified using the ExoSAP-IT™ (Applied Biosystems, Foster City, CA) PCR Product clean-up kit. Forward and reverse sequence reaction was prepared using the BigDye® Terminator Kit v3.1 (Applied Biosystems, Foster City, CA). Sequencing products were cleaned and precipitated using ethanol and NaAC and sequenced using an ABI Prism™ 3100 Automated Capillary DNA sequencer (Applied Biosystems) at the Bioinformatics Sequencing facility of the University of Pretoria (South Africa).

## Population genetic analyses

Electropherograms for all sequences were visualised and a consensus sequence generated using the Biological Sequence Alignment Editor (BioEdit) software (version 7.0.5) [74] and aligned using the online software Multiple Alignment using Fast Fourier Transform (MAFFT) v.7 [75]. To account for different sequence lengths, sequences were trimmed at 642 bp for the COI gene and 443 bp for the Cytb gene. Sequences of both COI and Cytb were concatenated to yield a total length of 1085 bp. Sequences of COI (accession numbers: OM263477-OM263633), and Cytb (accession numbers: OM219650-OM219806) of *B. distincta* were deposited in NCBI GenBank (www.ncbi.nlm.nih.gov). All the COI sequences obtained in this study were also submitted in the BOLD database (http://boldsystems.org/) under the project "PBDSA-*Bathycoelia distincta* Pentatomidae South Africa" (ID numbers: PMSL01-66 sequences from Limpopo; PMSM01-45 sequences from Mpumalanga; PMSK01-46 sequences from Kwazulu-Natal).

The COI and Cytb haplotype networks were constructed using Population Analysis with Reticulate Trees (PopART) version 1.7 [76]. Uncorrected (p) pairwise mean genetic distances between populations were calculated using the Kimura 2-parameter substitution model with 1000 bootstraps replicated in MEGA version 7.0.21 [77]. Genetic diversity parameters were determined using DnaSP 5.10.01 [78] and included the number of haplotypes (h), haplotype diversity (Hd), nucleotide diversity (*Pi*), genetic differentiation (Fst) and gene flow (Nm) values. The levels of genetic differentiation can be categorized as $F_{ST} > 0.25$ (high differentiation), 0.15 to 0.25 (moderate differentiation), and $F_{ST} < 0.05$ (negligible differentiation) [79]. The levels of gene flow can be categorized as Nm >1 (high gene flow), 0.25 to 0.99 (intermediate gene flow), and Nm <0.25 (low gene flow). Tajima's D [80] and Fu's Fs [81] values were estimated to test for changes in population size of *B. distincta* using DnaSP (v5.10.01). Significant negative values generally suggest population expansion. One thousand simulations under a model of selective neutrality were used to generate Tajima's D and Fu's Fs values. To determine the occurrence of isolation by distance (IBD), Mantel tests between the genetic and geographic distances between each population and marker were conducted using GenAlEx 6.5 with 9999 permutations [82].

## Results

### Sequence variation

The final sequence aligned matrix of the COI gene from 157 individuals was 642 bp in length. Genetic diversity indices for the COI gene are shown in Table 3. A total of 40 polymorphic nucleotides were observed. Thirty-five haplotypes were identified amongst COI sequences. The farms L1 and L2 showed 14 and 13 haplotypes from a total of 38 and 28 samples, respectively. Only 11 and 8 haplotypes were obtained from 45 and 46 samples of the farms M1 and K1, respectively. The haplotype diversity was the highest in Limpopo (farm L2, Hd = 0.754) and the lowest in the Kwazulu-Natal (farm K1, Hd = 0.571). The estimated nucleotide diversity

**Table 3. Summary of molecular diversity indices and population expansion test statistics of COI, Cytb and COI+Cytb genes in *Bathycoelia distincta* populations.**

| Gene | | n | h | S | k | Hd | *Pi* | D | F$_S$ |
|---|---|---|---|---|---|---|---|---|---|
| COI | Population | | | | | | | | |
| | Farm L1 | 38 | 14 | 15 | 0.989 | 0.606 | 0.00154 | -2.32655** | -13.391** |
| | Farm L2 | 28 | 13 | 18 | 1.677 | 0.754 | 0.00261 | -2.22585** | -8.615* |
| | Farm M1 | 45 | 11 | 13 | 1.285 | 0.725 | 0.00200 | -1.73408 | -5.511* |
| | Farm K1 | 46 | 8 | 11 | 0.899 | 0.571 | 0.00140 | -1.89467* | -3.600** |
| **All samples** | | **157** | **35** | **40** | **1.316** | **0.739** | **0.00206** | **-2.46755**** | **-43.196**** |
| Cytb | Population | | | | | | | | |
| | Farm L1 | 38 | 14 | 13 | 2.004 | 0.782 | 0.00452 | -1.10959 | -7.080 |
| | Farm L2 | 28 | 11 | 8 | 1.587 | 0.746 | 0.00358 | -0.99040 | -5.860 |
| | Farm M1 | 45 | 10 | 12 | 0.786 | 0.431 | 0.00177 | -2.15056* | -7.174* |
| | Farm K1 | 46 | 7 | 9 | 0.391 | 0.246 | 0.00088 | -2.30002** | -5.923** |
| **All samples** | | **157** | **34** | **30** | **1.174** | **0.549** | **0.00265** | **-2.31555**** | **-33.271**** |
| COI+Cytb | Population | | | | | | | | |
| | Farm L1 | 38 | 23 | 28 | 2.993 | 0.936 | 0.00276 | -1.90073* | -18.195* |
| | Farm L2 | 28 | 21 | 26 | 3.265 | 0.955 | 0.00301 | -1.92193* | -17.779* |
| | Farm M1 | 45 | 19 | 25 | 2.071 | 0.831 | 0.00191 | -2.11947* | -13.579* |
| | Farm K1 | 46 | 12 | 20 | 1.290 | 0.643 | 0.00119 | -2.31566** | -6.770** |
| **All samples** | | **157** | **62** | **70** | **2.490** | **0.875** | **0.00229** | **-2.52468***** | **-86.981**** |

For each population and gene marker, the number of specimens (n), number of haplotype (h), number of polymorphic (segregation) sites (S), average number of nucleotide differences (k), haplotype diversity (Hd), nucleotide diversity (pi) and Tajima's D (D) and Fu's Fs (F$_S$) tests statistics are given. Values are significant at

\* $P \leq 0.05$;

\*\* $P \leq 0.01$;

\*\*\* $P \leq 0.001$.

(*Pi*) was low overall, ranging from 0.00140 to 0.00261. When all samples were included, the total diversity was Hd = 0.739, but the nucleotide diversity was quite low *Pi* = 0.00206.

Genetic diversity indices for the Cytb gene are shown in Table 3. The final aligned sequence was composed of 443 nucleotides for the Cytb with a total of 30 polymorphic nucleotides observed. The total number of haplotypes was 34. The farm L1 revealed the highest number of haplotypes (14 haplotypes for 38 samples), while the farm K1 showed the lowest number of haplotypes (only 7 haplotypes for 46 samples). The total haplotype diversity was Hd = 0.549, ranging from 0.246 to 0.782. Low nucleotide diversity (*Pi*) was observed (Pi < 0.01). It was the highest in Limpopo (farm L1, *Pi* = 0.00452) and lowest in Kwazulu-Natal (farm K1, *Pi* = 0.00088).

The results of the combined analysis of both COI and Cytb genes are presented in Table 3. The 1085 paired bases analysed revealed 70 polymorphic nucleotides. In total, 62 haplotypes were obtained with haplotype diversity Hd = 0.875 and a nucleotide diversity *Pi* = 0.00229. The highest number of haplotypes was observed in Limpopo at the farm L1 with 23 haplotypes, while the lowest number was observed in Kwazulu-Natal with 12 haplotypes. However, the highest haplotype diversity and nucleotide diversity was observed in Limpopo at the farm L2 (Hd = 0.955, *Pi* = 0.00301). It was the lowest in Kwazulu-Natal (Hd = 0.643 and *Pi* = 0.00119).

The pairwise distance comparison among *B. distincta* populations based on COI, Cytb and both COI+Cytb genes are shown in Table 4. Sequence divergence among the four populations by pairwise comparison ranged from 0.2–0.4%. The highest sequence divergence was found

**Table 4. Uncorrected "p" distance matrix between different locations based on COI, Cytb and COI+Cytb DNA sequences of *Bathycoelia distincta* from South Africa.**

| COI | Farm L1 | Farm L2 | Farm M1 | Farm K1 |
|---|---|---|---|---|
| Farm L1 | - | | | |
| Farm L2 | 0.002 | - | | |
| Farm M1 | 0.002 | 0.003 | - | |
| Farm K1 | 0.002 | 0.003 | 0.002 | - |
| **Cytb** | Farm L1 | Farm L2 | Farm M1 | Farm K1 |
| Farm L1 | - | | | |
| Farm L2 | 0.004 | - | | |
| Farm M1 | 0.003 | 0.003 | - | |
| Farm K1 | 0.003 | 0.003 | 0.001 | - |
| **COI+Cytb** | Farm L1 | Farm L2 | Farm M1 | Farm K1 |
| Farm L1 | - | | | |
| Farm L2 | 0.003 | - | | |
| Farm M1 | 0.003 | 0.003 | - | |
| Farm K1 | 0.003 | 0.003 | 0.002 | - |

when farm L1-Cytb was compared with farm L2-Cytb. The lowest sequence divergence was observed between the farms M1-Cytb and K1-Cytb.

## Haplotype network analysis

To clarify the genetic relationship between *B. distincta* populations collected from various farms over different regions in South Africa, median-joining networks were generated (Figs 1–3). The obtained networks were generated with the genetic indices and neutrality tests calculated previously. The COI network (Fig 1) showed a star-like pattern with the two common haplotypes (Hap_C2 and Hap_C19) (S1 Table). The two common haplotypes included *B. distincta* collected from all the different farms in all three regions. They were separated by a single mutational change. For the Cytb, the generated haplotype network showed a dominant haplotype Hap_B1 where *B. distincta* collected from the four farms in all three regions are represented (Fig 2, S2 Table). Similar to the COI network, two common haplotypes Hap_CB3 and Hap_CB13 were found in the COI+Cytb median-joining network (Fig 3, S3 Table). The median-joining network of COI, Cytb, and COI+Cytb also showed a high number of unique haplotypes (35 for COI, 34 for Cytb, 62 for COI+Cytb), suggesting population expansions.

## Neutrality test

The neutrality test was conducted using Tajima's D and Fu's Fs statistics (Table 3). The results of the tests indicated significant negative D and $F_S$ values for the total of the populations for each gene (COI: D = -2.467, P < 0.01, $F_S$ = -43.196, P < 0.01; Cytb: D = -2.315, P < 0.01, $F_S$ = -33.271, P < 0.01; COI+Cytb: D = -2.525, P < 0.001, $F_S$ = -86.981, P < 0.01), suggesting population expansion. Considering each population separately, all the farms presented significant negative values for the neutrality tests, indicating that there is an excess of rare mutations in *B. distincta* populations which can imply recent population growth. In concordance, the haplotype networks indicated that the different sequence types observed in South Africa would have derived from a common ancestral haplotype (Hap_2).

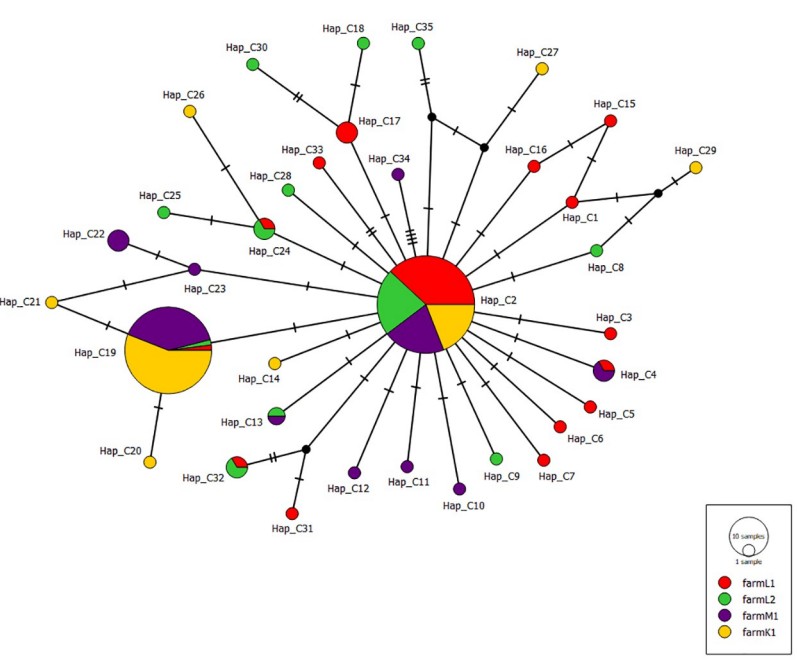

**Fig 1. Median-joining haplotype network of *Bathycoelia distincta* for COI gene.** Each haplotype is represented by a circle. Relative sizes of the circle indicate haplotype frequency. Color patterns demonstrate samples collected from different farms and regions of South Africa (Limpopo: farm L1 (n = 38), farm L2 (n = 28); Mpumalanga farm M1 (n = 45); Kwazulu-Natal farm K1 (n = 46)). Crossbars indicate one mutational step.

## Genetic structure

Genetic distance ($F_{ST}$) and migration rates (*Nm*) of the *B. distincta* populations were calculated (Table 5). Based on COI sequence data, the pairwise $F_{ST}$ among 4 pairs of *B. distincta* populations ranged from -0.005 to 0.291. *Bathycoelia distincta* samples from the Limpopo farms (L1 and L2) exhibited statistically significant genetic differentiation when compared to Mpumalanga (M1) and Kwazulu-Natal (K1) farms. Interestingly, similar results were obtained based on the COI+Cytb data set. Consistent results were also observed based on the Cytb dataset, which showed no significant differences ($P < 0.05$) of pairwise $F_{ST}$ in most population pairs except for the farms L1 and L2 when compared to the farm K1. Regarding the migration rates (*Nm*) values, all population pairs were greater than one, except for the Limpopo farms (COI, *Nm* = -48.96), or between the Mpumalanga and Kwazulu-Natal farms (Cytb, *Nm* = -90.55). Finally, for each marker, the Mantel test showed no statistically significant IBD, indicating no positive correlation between the geographic and genetic distances among *B. distincta* populations (COI: *r* = 0.731; P > 0.05; CytB: *r* = 0.310, P > 0.05; COI+Cytb: *r* = 0.669, P > 0.05) (S1–S3 Figs). Our results suggested that more than one stink bug female per generation was estimated to migrate between all pairs of populations, except between Limpopo and Mpumalanga or Kwazulu-Natal.

## Discussion

Population genetic studies have played a significant role in the identification of cryptic species and the study of their genetic diversity [42, 51]. In this study, we investigated the population genetics of *B. distincta* for the first time. DNA sequences were analysed employing COI and Cytb markers from specimens collected in the three main macadamia production regions of

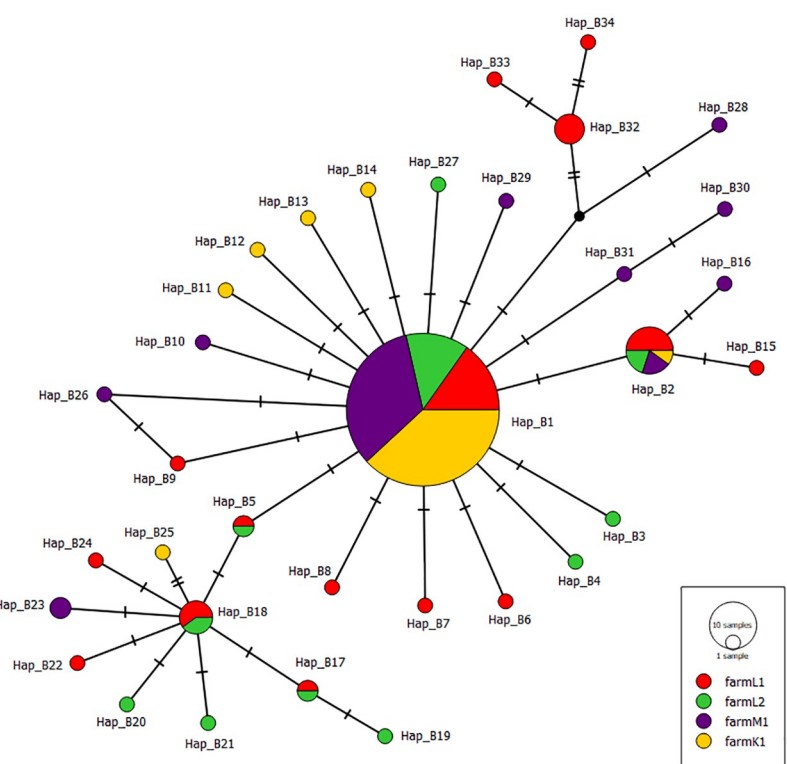

**Fig 2. Median-joining haplotype network of *Bathycoelia distincta* for Cytb gene.** Each haplotype is represented by a circle. Relative sizes of the circle indicate haplotype frequency. Color patterns demonstrate samples collected from different farms and regions of South Africa (Limpopo: farm L1 (n = 38), farm L2 (n = 28); Mpumalanga farm M1 (n = 45); Kwazulu-Natal farm K1 (n = 46)). Crossbars indicate one mutational step.

South Africa. The highest genetic diversity was observed within the Limpopo province populations, the area from where the species was first discovered. Pairwise mean distance analysis between populations suggested the absence of cryptic species. The median joining network for both datasets consisted of a few central haplotypes shared among populations in South Africa, with several unique haplotypes (35 for COI, 34 for Cytb, 62 for COI+Cytb) among the 157 *B. distincta* individuals examined (Figs 1–3). Considering the direct relationship between haplotype frequency and the ages of haplotypes [83, 84], the existence of a star-like structure (central haplotype and several less frequently derived haplotypes) suggests that most of the haplotypes originated recently, and is indicative of a population expansion during the recent history of the species [85].

   *Bathycoelia distincta* was originally discovered from the Limpopo region [14], and became a serious problem in South African macadamia areas in Limpopo and Mpumalanga in the early 2000s [10, 15, 86] and more recently in Kwazulu-Natal. The rapid and intensive increase in macadamia planted area in the last decade may be related to the genetic diversity observed in this study. Beck and Reese [87] proposed that insect survivorship, fecundity, growth rate and activity can be affected by the quantity and quality of the host. An abundance of hosts in a habitat will increase survival and fecundity and reduce mortality. As macadamia trees are the primary cultivated host for *B. distincta*, the increase in population size of this host would distinctly improve the fecundity and survival of the insect, and thus the extensive commercial

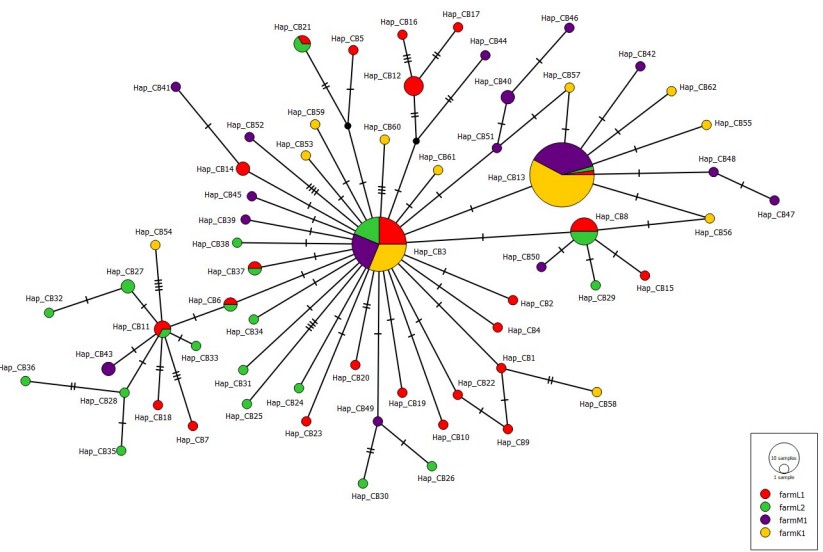

**Fig 3. Median-joining haplotype network of *Bathycoelia distincta* for COI+Cytb genes.** Each haplotype is represented by a circle. Relative sizes of the circle indicate haplotype frequency. Color patterns demonstrate samples collected from different farms and regions of South Africa (Limpopo: farm L1 (n = 38), farm L2 (n = 28); Mpumalanga farm M1 (n = 45); Kwazulu-Natal farm K1 (n = 46)). Crossbars indicate one mutational step.

cultivation of macadamia trees in recent years may have contributed to an expansion of *B. distincta*'s range in South Africa. Similar results have been obtained in *E. heros* populations in Brazil after the rapid expansion of the soybean-planted area, leading to an absence of genetic differentiation within populations [68], compared to the results obtained in the same area 14 years earlier [69]. This hypothesis is reinforced by the significant negative values found for the

**Table 5. Genetic differentiation (F$_{ST}$) and gene flow (*Nm*) based on COI, Cytb and COI+Cytb DNA sequences of *Bathycoelia distincta* from three different regions of South Africa.**

| COI | Farm L1 | Farm L2 | Farm M1 | Farm K1 |
|---|---|---|---|---|
| **Farm L1** | - | -48.96 | 1.58 | 0.61 |
| **Farm L2** | -0.00513 | - | 2.01 | 0.84 |
| **Farm M1** | 0.13647** | 0.11052** | - | 7.05 |
| **Farm K1** | 0.29150*** | 0.23025*** | 0.03427 | - |
| *Cytb* | **Farm L1** | **Farm L2** | **Farm M1** | **Farm K1** |
| **Farm L1** | - | 5.07 | 4.44 | 2.75 |
| **Farm L2** | 0.04703 | - | 2.21 | 1.44 |
| **Farm M1** | 0.05335 | 0.10151 | - | -90.55 |
| **Farm K1** | 0.08331* | 0.14768* | -0.00277 | - |
| *COI+Cytb* | **Farm L1** | **Farm L2** | **Farm M1** | **Farm K1** |
| **Farm L1** | - | 9.56 | 2.45 | 1.08 |
| **Farm L2** | 0.02548 | - | 2.10 | 1.02 |
| **Farm M1** | 0.09256* | 0.10654 | - | 11.32 |
| **Farm K1** | 0.18840* | 0.19643** | 0.02161 | - |

The data above and below the diagonal correspond to *Nm* and F$_{ST}$ respectively.

*, ** and *** indicate significant difference at P<0.05, P< 0.01 and P<0.001 respectively.

neutrality tests (Tajima's D and Fu's Fs), confirming a recent population expansion of *B. distincta*.

In our study, high gene flow ($Nm > 1$), lack of genetic differentiation ($F_{ST} < 0.1$) and no IBD ($P > 0.05$) were observed among the different populations (Table 5, S1–S3 Figs). Genetic differentiation between insect populations can be influenced by different factors such as the geographic distance between populations [53, 88, 89]. Although population differentiation by distance has been observed previously within Pentatomidae species [53, 57, 69], our results suggest that geographic distances did not affect the genetic structure of *B. distincta*. While limited genetic structure and high levels of population connectivity were expected for the farms located in Limpopo (distance between farms L1-L2 is 10 km), this was not expected for our regional scale investigation as the farms sampled are geographically distant, being up to 800 km apart (Table 2). Few studies have been conducted on the flight capacities of Pentatomidae using flight mills. Babu et al. [90] showed that *Euschistus servus* (Say) can fly a maximum distance of 15.9 km in 22 h, especially after overwintering emergence, whereas most individuals only flew between 0–1 km. For *H. halys*, adults can fly 5–7 km in 24 h and up to 117 km, with longer and faster flights achieved in summer [91–93], while nymphs can easily walk among host plants [94]. Considering that genetic isolation can also be higher in species with a limited capacity for dispersal [95], a high dispersal capacity in *B. distincta* could explain the genetic homogeneity among its populations in South Africa.

Factors other than flight ability might also facilitate movement of *B. distincta*. One such a factor is the distribution of suitable host plants (both wild and cultivated) in South Africa. Stink bugs can feed on several plant-hosts [2, 96] and higher levels of damage are often observed when forests and natural vegetation border crops [97, 98], a feature quite common in the South African landscape. In addition, the two-spotted stink bug population can reach high densities especially in the canopies [20], mainly from November to March when nuts are present but they can remain in the field even after the nuts have been harvested [10, 14]. After harvest *B. distincta* may disperse, looking for shelter to remain in diapause during winter. High levels of gene flow, as determined for *B. distincta* in this study, are determinants of the maintenance of high levels of genetic diversity and low population differentiation [99, 100]. This, in addition to its multivoltinism, may have contributed to high population densities, a wide distribution, and low levels of population differentiation throughout the country. Therefore, these long-distance dispersal events might occur in parallel with some anthropogenic driven dispersal (e.g., transport of seedlings or fruits among regions of South Africa).

Genetic divergence and nucleotide diversity in this study among the different locations was comparable to results for Hemiptera in previous studies. The intraspecific genetic divergence in this study was in the range of 0.2 to 0.4% (Table 4) and a low nucleotide diversity was observed ($Pi < 3\%$) (Table 3). Previous studies observed intraspecific divergence of 4.7% between individuals of *C. hilaris* [53], and > 2% in *N. viridula* [54]. Nevertheless, an intraspecific genetic divergence value of 4.7% has been suggested to delineate cryptic species within populations of Heteroptera [51]. Considering the previous results obtained and the morphological and ecological similarity of *B. distincta* between the different locations analysed, we can confirm that our study did not show any evidence of cryptic species in its populations in South Africa. Knowledge of the existence of cryptic species is important for IPM because different species can respond differently to pest management strategies [23, 35]. For example, variation in the pheromone blends have been found among different populations of *Diatraea saccharalis* (Fabricius), suggesting that the trapping efficiency could vary among the regions [26]. Similarly, insect populations from different geographic regions can vary in their level of susceptibility to insecticides as it has been recently demonstrated in wireworm populations [101]. Thus,

it would be useful to determine the effectiveness of future pheromone lures and other control methods across the wide geographical range of *B. distincta*.

In conclusion, this study of *B. distincta* collected within the three main macadamia production regions in South Africa revealed a high genotypic diversity and a lack of genetic differentiation among localities. The high genotypic diversity suggests favourable environmental conditions for reproduction and growth of the species in its native range. The high gene flow observed, even across a wide geographic area, appears to be the major ecological force shaping the overall genetic pattern observed. Regional population dynamics of *B. distincta* are likely linked to the rapid expansion of macadamia planted areas. To our knowledge this is the first study on the population genetics of *B. distincta*. This information provides a critical starting point for understanding this species in South Africa and might assist future development of pest management strategies that incorporate pheromones and biological control. Furthermore, future studies using other genetic and genomic tools and including a larger sample size and geographic range, could help understand the movement of on *B. distincta* populations between orchards and native plants, and within regions and over shorter time scales. Such genetic and genomic tools are currently under-utilized as a resource for stink bug management and our study provides a foundation for such further work.

## Supporting information

**S1 Fig. Isolation by distance of *Bathycoelia distincta* populations for COI marker (Mantel test, *r* = 0.731, P > 0.05).**
(TIF)

**S2 Fig. Isolation by distance of *Bathycoelia distincta* populations for Cytb marker (Mantel test, *r* = 0.310, P > 0.05).**
(TIF)

**S3 Fig. Isolation by distance of *Bathycoelia distincta* populations for COI+Cytb combined marker (Mantel test, *r* = 0.669, P > 0.05).**
(TIF)

**S1 Table. List of the individual for each haplotype generated in the study for the COI marker.**
(DOCX)

**S2 Table. List of the individual for each haplotype generated in the study for the Cytb marker.**
(DOCX)

**S3 Table. List of the individual for each haplotype generated in the study for the COI +Cytb combined markers.**
(DOCX)

## Acknowledgments

This study was made possible by the Forestry and Agricultural Biotechnology Institute at the University of Pretoria, who provided the facilities. We are grateful to farmers and landowners in the Limpopo, Mpumalanga and Kwazulu-Natal for access and sample collection.

## Author Contributions

**Conceptualization:** Elisa Pal, Jeremy D. Allison, Brett P. Hurley, Bernard Slippers, Gerda Fourie.

**Data curation:** Elisa Pal, Jeremy D. Allison, Brett P. Hurley, Gerda Fourie.

**Formal analysis:** Elisa Pal.

**Funding acquisition:** Gerda Fourie.

**Investigation:** Elisa Pal.

**Methodology:** Elisa Pal.

**Resources:** Brett P. Hurley, Bernard Slippers.

**Software:** Elisa Pal.

**Supervision:** Jeremy D. Allison, Brett P. Hurley, Bernard Slippers, Gerda Fourie.

**Validation:** Elisa Pal, Jeremy D. Allison, Bernard Slippers, Gerda Fourie.

**Visualization:** Elisa Pal, Jeremy D. Allison, Bernard Slippers, Gerda Fourie.

**Writing – original draft:** Elisa Pal.

**Writing – review & editing:** Elisa Pal, Jeremy D. Allison, Brett P. Hurley, Bernard Slippers, Gerda Fourie.

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
