## [Decision Letter · Decision Letter 0]

7 Apr 2022

PONE-D-22-02292Genetic diversity of the two-spotted stink bug Bathycoelia distincta (Pentatomidae) associated with macadamia orchards in South AfricaPLOS ONE

Dear Dr. Fourie,

Thank you for submitting your manuscript to PLOS ONE. After careful consideration, we feel that it has merit but does not fully meet PLOS ONE’s publication criteria as it currently stands. Therefore, we invite you to submit a revised version of the manuscript that addresses the points raised during the review process.

We look forward to receiving your revised manuscript.

Kind regards,

Patrizia Falabella

Academic Editor

PLOS ONE

Journal Requirements:

(We would like to acknowledge the University of Pretoria, the Forestry and Agricultural Biotechnology Institute (FABI), the Centre for excellence in Plant Health Biotechnology (CPHB), Macadamias South Africa NPC (SAMAC) and NRF Thuthuka for financial support)

5. We note that Figure 1 in your submission contain map images which may be copyrighted. All PLOS content is published under the Creative Commons Attribution License (CC BY 4.0), which means that the manuscript, images, and Supporting Information files will be freely available online, and any third party is permitted to access, download, copy, distribute, and use these materials in any way, even commercially, with proper attribution. For these reasons, we cannot publish previously copyrighted maps or satellite images created using proprietary data, such as Google software (Google Maps, Street View, and Earth). For more information, see our copyright guidelines: http://journals.plos.org/plosone/s/licenses-and-copyright.

Reviewers' comments:

Reviewer's Responses to Questions

**Comments to the Author**

1. Is the manuscript technically sound, and do the data support the conclusions?

Reviewer #1: Yes

Reviewer #2: No

Reviewer #3: Yes

2. Has the statistical analysis been performed appropriately and rigorously? 

Reviewer #1: Yes

Reviewer #2: Yes

Reviewer #3: Yes

3. Have the authors made all data underlying the findings in their manuscript fully available?

Reviewer #1: Yes

Reviewer #2: No

Reviewer #3: Yes

4. Is the manuscript presented in an intelligible fashion and written in standard English?

Reviewer #1: Yes

Reviewer #2: Yes

Reviewer #3: Yes

5. Review Comments to the Author

Reviewer #1: The manuscript entitled "Genetic diversity of the two-spotted stink bug Bathycoelia distincta (Pentatomidae) associated with macadamia orchards in South Africa" by Fourie et al described the genetic diversity of B. distincta populations in South Africa using COI and Ctyb gene markers. The population analysis using different methods have been conducted using the two markers separately and combined. It is a nice study and the manuscript provided novel information on the two-spotted stink bug. I have a couple of suggestions as below:

1) The information on how to identify the sting bug is lacking in the manuscript. The method used for the morphological identification should be added, such as which morphological keys were used and identified by whom…

2) Suggest the authors to submit the COI sequence to BOLD databases as there are no COI sequence available in the databases until now.

3) For the name of the haplotype, similar names were used for COI, cytb and COI-cytb combined. I think it will be clearer if the authors can use slightly different names to distinct them, for example, Hap_C1 for COI, Hap_B1 for cytB, Hap_CB1 for combined (Note: just my suggested name systems, do not need to follow my name conventions)? In this way, it will be easy for readers to follow when Hap_B1 mentioned, they will know it is talking about cytB haplotype.

Minor comments:

1. Table 1, Table 2 and Figure1 are presenting the same information, it seems that figure 1 is not necessary.

2. Lines 159-160: suggest providing the primer names for the two cytb primers.

3. Lines 157-172: the authors used two different PCR mastermixes to amplify the COI and cytb genes. Just wondering the reasons for the authors not to use the same matermix for amplify the two genes. According to my experience, generally it will be able to amplify the two genes under the same PCR compositions.

4. Lines 177-181: the detail pressure on how to sequence is not necessary as nowadays, it is standard protocol for the sequencing provider to run the sequence.

5. Line 371: D. saccharalis mentioned in the first time, the genus name should be used.

.Reviewer #2: This paper looks at population genetics of Bathycoelia distincta. It is well-written and concise. However, I have a few concerns regarding the work described. These need clarification before the paper can be fully accepted.

Regarding the M&M, the authors appear to not include any controls. This include several insect outgroups (other Heteroptera) as well as several Pentatomids other than those in the same Family. I would also include several within the same Family. Secondly, the sequencing for each individual must be replicated a minimum of three times (forward and reverse) for each individual and each primer set. Failure to do so can result in misalignment and mis-sequencing. Finally, no indication is given as to where the insects tested are stored or the voucher specimens are located. Thus, it appears from first reading that the experiments were not conducted as rigorously as they could have and the appropriate controls are missing.

Reviewer #3: "Genetic diversity of the two-spotted stink bug Bathycoelia distincta (Pentatomidae) associated with macadamia orchards in South Africa"

Overall, this paper is very well written, and makes an important contribution to a newly emerging insect pest of macadamia in South Africa.

There are a few very minor grammar suggestions.

line 64 edit 'damaging pest' to 'damaging pests'

line 76 remove the , after dynamics

Methods:

I have one major comment. The Fst values between K and L population are highest. These populations are geographically most distant. I suggest running a Mantel test, to examine the association between geographic distance and genetic divergence. It looks to be significant. This would be expected. Genalex can be used to run this test

Discussion: Line 379. Why do the authors believe there are so many new haplotypes? The star pattern in the haplotype tree shows they all branch off one or two main haplotype. What might cause many new mutations and haplotypes? Insecticides?

Some discussion of why there are so many new haplotypes would add to the manuscript.

6. PLOS authors have the option to publish the peer review history of their article (what does this mean?). If published, this will include your full peer review and any attached files.

Reviewer #1: **Yes: **Dongmei Li

Reviewer #2: No

Reviewer #3: No

---

## [Author Response · Author response to Decision Letter 0]

25 Apr 2022

Dr Patrizia Falabella

Academic Editor

PLOS ONE

Submission of Revised Manuscript

Thank you for your positive feedback regarding our manuscript, entitled “Genetic diversity of the two-spotted stink bug Bathycoelia distincta (Pentatomidae) associated with macadamia orchards in South Africa”. We believe that the suggestions made by the reviewers have improved the scientific quality and clarity of our manuscript. 

In this revision, we incorporated all the suggestions made by the reviewers, which has resulted in a number of minor changes to our manuscript. 

Please find attached our revised manuscript. Our response to the comments and suggestions made by the reviewers can be found in the document below titled “Response to Reviewers”, a marked copy of the manuscript can be found in the document titled “Revised Manuscript with Track Changes” and the unmarked version of the manuscript can be found in the document titled “Manuscript”. 

We thank you in advance for your assistance in handling this manuscript further.

Kindest regards,

Journal requirements

Please note that the page and line numbers referred to in our responses represent those found in the newly submitted manuscript. Furthermore, specific changes are indicated in italics, where appropriate.

1) Please ensure that your manuscript meets PLOS ONE’s style requirements, including those for file naming.

• The manuscript and file names of the manuscript was reviewed in order to ensure that all the PLOS ONE’s style requirements are met. 

2) In your methods section, please provide additional information regarding the permits you obtained for the work.

• No permit is needed in South Africa. An additional section has been added Lines 130-132: “Ethics statement. No endangered or protected species were involved in this study. No national permissions were required for this study. All work on this project was conducted with permission from landowners”.

3) Funding statement modifications.

• The funding statement was updated online and in the cover letter.

4) We note that the grant information you provide in the “funding information” and “Financial Disclosure” sections do not match.

• As mentioned above, funding information was updated and corrected. 

5) We note that Figure 1 in your submission contain map images which may be copyrighted.

• As suggested by the reviewer 1, Figure 1 was removed from the manuscript.

Response to Reviewers

Please note that the page and line numbers referred to in our responses represent those found in the newly submitted manuscript. Furthermore, specific changes are indicated in italics, where appropriate.

Reviewer 1: 

1) The reviewer suggested that information on how to identify the stink bug species was lacking and that the method for morphological identification and morphological keys should be added. 

• Lines 140-146: We included more detail about the method for morphological identification by adding, “The two-spotted stink bug specimens were identified based on external morphological features described in literature [72] and also confirmed morphologically by an entomologist at the Agricultural Research council – Plant Health and Protection (Pretoria, South Africa). Insects used for this study are conserved in our collection located at University of Pretoria. Vouchered specimens were pinned, accession number assigned (PENT00026-PENT00030) and deposited in the National Collection of Insects located at the Agricultural Research Council – Plant Health and Protection (Pretoria, South Africa).” 

2) The reviewer suggested to submit our COI sequences to BOLD database.

• Our COI sequences were deposited to BOLD database under accession numbers PMSL01-66 sequences from Limpopo, PMSM01-45 sequences from Mpumalanga, PMSK01-46 sequences from Kwazulu-Natal. This was also stated in the revised manuscript Lines 196-199.

3) The reviewer suggested to use different names to distinguish between haplotypes.

• This is a valid suggestion. We have revised the names used for each haplotype into “Hap_C1” for COI, “Hap_B1” for Cytb and “Hap_CB1” for combined COI-Cytb as suggested.

4) The reviewer suggested a number of minor comments.

1. Table 1, Table 2 and Figure 1 are presenting the same information, it seems that figure 1 is not necessary: Figure 1 has been removed.

2. Lines 159-160, the reviewer suggests to provide the primer names for the two cytb primers: The two Cytb primers used in this study were developed by Muraji et al. 2000 and used in the study of Li et al. 2014. In both studies, no primer names were listed.

3. Lines 157-172 the authors used two different PCR mastermixes: Our initial tests of CytB amplification using a standard Taq DNA polymerase resulted in non-specific amplification. We therefore switched to using a FastStart Taq to increase the specificity. This was also added in the revised manuscript Line 176.

4. Lines 177-181 the reviewer suggested that the details for the sequencing was not needed: We removed the following from the manuscript “using 5.8 µL distilled water, 1 µL Big Dye, 1 µL sequencing buffer, 0.2 µL primer and 2 µL purified PCR product. Cycle sequencing conditions included initial denaturation of 2 min at 96°C followed by 25 cycles of 10 s at 96°C, 5 s at 50°C and 4 min at 60°C”.

5. Line 371 D. saccharalis mentioned for the first time: We are grateful to the reviewer for picking up this error. We have changed the species name to “Diatraea saccharalis” (line 394).

Reviewer 2:

1) The reviewer asked to include controls in the materials and methods such as several insect outgroups (other Heteroptera) as well as several Pentatomids other than those in the same family.

• We included more details in line 180: “With each run, negative and positive PCR controls were performed for PCR validation.”

• This is a valid point, however this study was focused specifically on the genetic diversity of

Bathycoelia distincta in South Africa and. It is our opinion that it was is not necessary to include an outgroup. For example, similar studies, published in PLoS ONE, such as Karsten et al. 2013 and Low et al. 2014 (references cited below) which were also focused on one species, did not include outgroups.

- Karsten M, van Vuuren BJ, Barnaud A, Terblanche JS (2013) Population Genetics of Ceratitis capitata in South Africa: Implications for Dispersal and Pest Management. PLoS ONE 8(1): e54281. doi:10.1371/journal.pone.0054281

- Low VL, Adler PH, Takaoka H, Ya’cob Z, Lim PE, et al. (2014) Mitochondrial DNA Markers Reveal High Genetic Diversity but Low Genetic Differentiation in the Black Fly Simulium tani Takaoka & Davies along an Elevational Gradient in Malaysia. PLoS ONE 9(6): e100512. doi:10.1371/journal.pone.0100512

2) The reviewer specified that the sequencing must be replicated a minimum of three times (forward and reverse) for each individual and each primer set.

• Each individual was sequenced in both directions (i.e. forward and reverse primer) using an ABI Prism™ 3100 Automated Capillary DNA sequencer (Applied Biosystems). The electropherograms were visualised, base calling accuracies were checked and a consensus sequence was generated. This type of sequencing provides highly accurate results (99.99%) compared to other next generation sequencing technologies such as Illumina and/or PacBio. Sequencing each sample with each individual primer set is not compulsory in this type of study. In addition, recent studies published in PLOS ONE journal this year (references below) did not repeat the sequencing of the gene markers COI and Cytb three times. For these reasons, we did not repeat our sequencing.

• The following sentence was added in the revised manuscript line 190: “Electropherograms for all sequences were visualised and a consensus sequence generated”.

Tsoupas A, Papavasileiou S, Minoudi S, Gkagkavouzis K, Petriki O, Bobori D, et al. (2022) DNA barcoding identification of Greek freshwater fishes. PLoS ONE 17(1): e0263118. https://doi.org/10.1371/journal.pone.0263118

Arnaout Y, Djelouadji Z, Robardet E, Cappelle J, Cliquet F, Touzalin F, et al. (2022) Genetic identification of bat species for pathogen surveillance across France. PLoS ONE 17(1): e0261344. https://doi.org/10.1371/journal.pone.0261344

3) The reviewer noted that no indication is given as to where insects tested are stored or the voucher specimens are located.

• The insects used in our study are conserved in our facility and some of the specimens were sent to a taxonomist at the Agricultural Research Council (ARC) in Pretoria for identification. Four of our specimens was deposited in the National Collection of Insects under the voucher numbers PENT00026-PENT00030.

• We added additional text Lines 140-146: “The two-spotted stink bug specimens were identified based on external morphological features described in literature [72] and also confirmed morphologically by an entomologist at the Agricultural Research Council – Plant Health and Protection (Pretoria, South Africa). Insects used for this study are conserved in our collection located at University of Pretoria. Vouchered specimens were pinned, accession number assigned (PENT00026-PENT00030) and deposited in the National Collection of Insects located at the Agricultural Research Council – Plant Health and Protection (Pretoria, South Africa).”

Reviewer 3:

1) The reviewer noted a few grammatical errors.

• Line 62: we corrected “damaging pest” to “damaging pests”

• Line 74: we removed “,” after dynamics

2) The reviewer suggested to conduct a Mandel test to examine the association between geographic distance and genetic divergence.

• Mantel tests were conducted and revealed no significant differences for each marker.

• The regression figures between the genetic and geographical distance among the populations for each genetic marker were added as supporting information: S1_Fig for COI, S2_Fig for Cytb and S3_Fig for COI+Cytb.

• Additional information related to these results were added in the manuscript:

o Lines 39-40: ….”absence of correlation between genetic and geographic distance”….

o Lines 212-214: “To determine the occurrence of isolation by distance (IBD), Mantel tests between the genetic and geographic distances between each population and marker were conducted using GenAlEx 6.5 with 9999 permutations [83]” 

o Line 300: the title of the paragraph was renamed “Genetic structure”

o Lines 310-313: “Finally, for each marker, the Mantel test showed no statistically significant IBD, indicating no positive correlation between the geographic and genetic distances among B. distincta populations (COI: r = 0.731; P > 0.05; CytB: r = 0.310, P > 0.05; COI+Cytb: r = 0.669, P > 0.05) (S1-S3 Figs). (COI: r = 0.731; P > 0.05; CytB: r = 0.310, P > 0.05; COI+Cytb: r = 0.669, P > 0.05) (S1-S3 Figs)”.

o Line 353: …”and no IBD (P > 0.05)”….

o Line 354: …”S1-S3 Figs.”

3) The reviewer addressed some questions regarding line 379 - “what might cause many new mutations and haplotypes? Insecticides?” - and suggested adding some discussion to the manuscript.

• We believe that the presence of many haplotypes is linked to the species being in its native range. The species is also currently the most dominant and damaging pest of macadamia. Over time the species adapted to feed on macadamia and since nymphs are often present in orchards it suggests that macadamia orchards are a favourably habitat for reproduction and growth. Although insecticides could also likely explain some of the many haplotypes present, no information regarding insecticide resistance is currently available and we therefore did not comment on this in the manuscript

• The following sentence was added in the revised manuscript Lines 403-404: “The high genotypic diversity suggests favourable environmental conditions for reproduction and growth of the species in its native range.”

---

## [Decision Letter · Decision Letter 1]

12 May 2022

PONE-D-22-02292R1Genetic diversity of the two-spotted stink bug Bathycoelia distincta (Pentatomidae) associated with macadamia orchards in South AfricaPLOS ONE

Dear Dr. Fourie,

Thank you for submitting your manuscript to PLOS ONE. After careful consideration, we feel that it has merit but does not fully meet PLOS ONE’s publication criteria as it currently stands. Therefore, we invite you to submit a revised version of the manuscript that addresses the points raised during the review process.

We look forward to receiving your revised manuscript.

Kind regards,

Patrizia Falabella

Academic Editor

PLOS ONE

Journal Requirements:

Reviewers' comments:

Reviewer's Responses to Questions

**Comments to the Author**

1. If the authors have adequately addressed your comments raised in a previous round of review and you feel that this manuscript is now acceptable for publication, you may indicate that here to bypass the “Comments to the Author” section, enter your conflict of interest statement in the “Confidential to Editor” section, and submit your "Accept" recommendation.

Reviewer #1: All comments have been addressed

2. Is the manuscript technically sound, and do the data support the conclusions?

Reviewer #1: Yes

3. Has the statistical analysis been performed appropriately and rigorously? 

Reviewer #1: Yes

4. Have the authors made all data underlying the findings in their manuscript fully available?

Reviewer #1: No

5. Is the manuscript presented in an intelligible fashion and written in standard English?

Reviewer #1: Yes

6. Review Comments to the Author

Reviewer #1: the comments raised by me were address and the revised MS has been edited accordingly. I have two minor points need to be addressed before it can be published.

1. the sequence submitted in BOLD need to be publicly available before publication;

2. line 180 "With each PCR run, negative and positive PCR controls were performed for PCR validation"- what is the negative control? which species used, how about positive control? could the authors specify what DNA used for the controls? Was non-template control used for the PCR reactions?

7. PLOS authors have the option to publish the peer review history of their article (what does this mean?). If published, this will include your full peer review and any attached files.

Reviewer #1: No

---

## [Author Response · Author response to Decision Letter 1]

17 May 2022

Submission of Revised Manuscript

Thank you for your positive feedback regarding our manuscript, entitled “Genetic diversity of the two-spotted stink bug Bathycoelia distincta (Pentatomidae) associated with macadamia orchards in South Africa”. We believe that the suggestions made by the reviewers (both rounds) have improved the scientific quality and clarity of our manuscript. In this revision, we incorporated the suggestions made by reviewer 1, which has resulted in minor changes to our manuscript. 

Please find attached our revised manuscript. Our response to the comments and suggestions made by the reviewer can be found in the document below titled “Response to Reviewers”, a marked copy of the manuscript can be found in the document titled “Revised Manuscript with Track Changes” and the unmarked version of the manuscript can be found in the document titled “Manuscript”. 

We thank you in advance for your assistance in handling this manuscript further.

Journal Requirements:

Please review your reference list to ensure that it is complete and correct. If you have cited papers that have been retracted, please include the rationale for doing so in the manuscript text, or remove these references and replace them with relevant current references

We have carefully reviewed our references and noted that reference [9] “DALRRD. Ressource Centre of the Department of Agriculture, Land Reform and Rural Development 2020. Available from: https://www.dalrrd.gov.za/.” are no longer available on-line. This reference was subsequently removed from our manuscript. The reference was not replaced since the information are also captured in reference [10].

- L56: "[9]" has been removed and as such the numbers from [9] onwards changed. 

Finally, we noted a small type error in the SAMAC reference (previously numbered as [10] but [9] in the revised manuscript) and as such was updated to:

- L458-459: " SAMAC (South African Macadamia Association). Industry statistics on the South African Macadamia Industry 2021. 2021. Available online at: https://www.samac.org.za/industry-statistics/"

Response to Reviewers

Reviewer 1: 

The reviewer had two minor points:

1) The sequence submitted in BOLD need to be publicly available before publication: 

- We made the sequences publicly available as requested. You can retrieve the sequences under the project “PBDSA-Bathycoelia distincta Pentatomidae South Africa”.

2) Line 180 "With each PCR run, negative and positive PCR controls were performed for PCR validation"- what is the negative control? which species used, how about positive control? could the authors specify what DNA used for the controls? Was non-template control used for the PCR reactions? 

- The negative control used consisted of sample without DNA. This was to ensure that the master mix was not contaminated. This was also stated in the revised manuscript Lines 180-181. “negative controls without DNA in the PCR reaction”. We removed the mention of a positive control from the manuscript. The positive control was added to each run as an internal control. The positive control comprised of B. distincta DNA that previously amplified and was used as a control to ensure that all the required PCR reagents were added to the master mix correctly.

---

## [Editor Report · Decision Letter 2]

20 May 2022

Genetic diversity of the two-spotted stink bug Bathycoelia distincta (Pentatomidae) associated with macadamia orchards in South Africa

PONE-D-22-02292R2

Dear Dr. Gerda Fourie,

We’re pleased to inform you that your manuscript has been judged scientifically suitable for publication and will be formally accepted for publication once it meets all outstanding technical requirements.

Kind regards,

Patrizia Falabella

Academic Editor

PLOS ONE

Additional Editor Comments (optional):

I suggest to insert in the manuscript a brief sentence in which it has reported the availability of the sequence in the project “PBDSA-Bathycoelia distincta Pentatomidae South Africa", with a specific website link.
---

## [Editor Report · Acceptance letter]

2 Jun 2022

PONE-D-22-02292R2 

Genetic diversity of the two-spotted stink bug *Bathycoelia distincta* (Pentatomidae) associated with macadamia orchards in South Africa 

Dear Dr. Fourie:

I'm pleased to inform you that your manuscript has been deemed suitable for publication in PLOS ONE. Congratulations! Your manuscript is now with our production department. 

Kind regards, 

on behalf of

Prof. Patrizia Falabella 

Academic Editor

PLOS ONE